

# Network intrusion detection using oversampling technique and machine learning algorithms

Hafiza Anisa Ahmed, Anum Hameed and Narmeen Zakaria Bawany

Department of Computer Science and Software Engineering, Jinnah University for Women, Karachi, Sindh, Pakistan

## ABSTRACT

The expeditious growth of the World Wide Web and the rampant flow of network traffic have resulted in a continuous increase of network security threats. Cyber attackers seek to exploit vulnerabilities in network architecture to steal valuable information or disrupt computer resources. Network Intrusion Detection System (NIDS) is used to effectively detect various attacks, thus providing timely protection to network resources from these attacks. To implement NIDS, a stream of supervised and unsupervised machine learning approaches is applied to detect irregularities in network traffic and to address network security issues. Such NIDSs are trained using various datasets that include attack traces. However, due to the advancement in modern-day attacks, these systems are unable to detect the emerging threats. Therefore, NIDS needs to be trained and developed with a modern comprehensive dataset which contains contemporary common and attack activities. This paper presents a framework in which different machine learning classification schemes are employed to detect various types of network attack categories. Five machine learning algorithms: Random Forest, Decision Tree, Logistic Regression, K-Nearest Neighbors and Artificial Neural Networks, are used for attack detection. This study uses a dataset published by the University of New South Wales (UNSW-NB15), a relatively new dataset that contains a large amount of network traffic data with nine categories of network attacks. The results show that the classification models achieved the highest accuracy of 89.29% by applying the Random Forest algorithm. Further improvement in the accuracy of classification models is observed when Synthetic Minority Oversampling Technique (SMOTE) is applied to address the class imbalance problem. After applying the SMOTE, the Random Forest classifier showed an accuracy of 95.1% with 24 selected features from the Principal Component Analysis method.

## INTRODUCTION

In today's developed and interconnected world, the number of networks and data security breaches is increasing immensely. The reasons include the growth of network traffic and advances in technology that have led to the creation of newer types of attacks. As a result, the level of attack eventually increases (*Mikhail, Fossaceca & Iammartino, 2019*).

Corresponding author
Hafiza Anisa Ahmed,
hafizaanisaahmed@gmail.com

There exist numerous network security attacks in today's era and to timely detect these attacks, several NIDSs are being developed and deployed. These NIDSs are widely used to protect digital resources against attacks and intrusions on networks (*Zong, Chow & Susilo, 2018*; *Vidal & VidalMonge, 2019*). Intrusion detection systems use two different methods, that is, anomaly-based detection and signature-based detection (*Moustafa, Creech & Slay, 2017*; *Li et al., 2019*). In an anomaly detection system, the network traffic is monitored and critical network characteristics are continuously tracked and analyzed (*Habeeb et al., 2018*). It generates alerts if unusual or anomalous activity is detected. Whereas, in signature detection system, well-known patterns of attacks (known as signatures) are stored. The network packets are searched for those patterns (*Faker & Dogdu, 2019*). If a pattern is accurately matched, the system generates an alert regarding that malicious activity (*Azeez et al., 2020*).

Although various attack detection mechanisms are available, they are still not effective enough to detect and analyze intrusions or malicious activities (*Ahmed, Mahmood & Hu, 2016*). Typically, anomaly-based detection systems are developed using different machine learning techniques for predicting intrusions in networks. Research has been conducted in this regard using datasets such as KDDCUP99 (*Choudhary & Kesswani, 2020*), KDD98 (W. *Haider et al., 2017*) and NSL-KDD7 (*Rathore & Park, 2018*). However, due to the evolution of computer networks, these datasets are negatively affecting the results of NIDS (*Khraisat et al., 2019*). One of the factors influencing the results is the availability of modern-day attack data, as these datasets were created almost two decades ago. Consequently, due to revolution of network traffic, the traffic data available in those datasets is different from the existing modern-day traffic (*Moustafa & Slay, 2016*).

Improving the performance of existing NIDS requires modern and state-of-the-art datasets that are up-to-date. Therefore, a more efficient and more accurate evaluation of NIDS requires relatively new state-of-the-art datasets, including modern-day network's normal and attack activities. In this research, a framework has been developed for attack detection in a network using the UNSW-NB15 dataset (*Tama & Rhee, 2019*; *Aissa, Guerroumi & Derhab, 2020*). This dataset is more recent and includes new attacks. Previously, KDDCUP99, KDD98, and NSL-KDD7 were widely used for NIDS benchmarked datasets. However, these older datasets are not as useful for today's network traffic (*Viet et al., 2018*; *Kumar et al., 2020*). Although, a few researchers have used the new data set, the UNSW-NB15 dataset, to detect an attack, their work has been limited (*Bagui et al., 2019*).

The model proposed in this study uses the UNSW-NB15 (https://research.unsw.edu.au/projects/unsw-nb15-dataset) dataset and not only achieves better accuracy than previous research (*Kasongo & Sun, 2020*; *Kumar, Das & Sinha, 2021*) but also effectively detects all categories of attack. The work has been done using Python. Initially, pre-processing techniques were used on 80,000 randomly selected instances from the UNSW-NB15 dataset to normalize data values. Later, feature engineering has been performed to select the relevant features. To improve the performance of the classifiers, the research solved the problem of class imbalance using SMOTE. Subsequently, Random Forest (RF), Decision Tree (DT), Logistic Regression (LR), K-Nearest Neighbors (KNN) and Artificial Neural

Network (ANN) have been used for classification. Lastly, evaluation metrics were used to compare the performance of all classifiers.

Following are the major contributions of this research:

- The dataset includes 45 features from which we identified 24 features that were most significant in identifying the attack.
- The various pre-processing techniques have been collectively applied to the UNSW-NB15 dataset to make the data meaningful and informative for model training.
- The class imbalance problem is addressed using the Synthetic Minority Oversampling Technique (SMOTE), thereby improving the detection rate of rare attacks.
- We have provided a comparison of five machine learning algorithms for detecting network attack categories.

The rest of the paper is organized as follows: In "Related Work", related work has been presented; "Proposed Methodology" describes the methodology of the framework developed; "Experiment and Result Analysis" and "Discussion" elaborates the discussion of the experimental results and the last "Conclusions" concludes the paper.

## RELATED WORK

As technology advances with modern techniques, computer networks are using the latest technologies to put it into practice, which has dramatically changed the level of attacks. Therefore, to target the present-day attack categories, UNSW-NB15 dataset has been created (*Moustafa & Slay, 2015*; *Viet et al., 2018*).

The research conducted using the UNSW-NB15 dataset is still not sufficient. However, some of the research work done using datasets is discussed below. Table 1 presents the summary and comparison of the discussed related work.

*Moustafa & Slay (2015)* developed a model that focused on the classification of attack families available in the UNSW-NB15 dataset. The study used the Association Rule Mining technique for feature selection. For classification, Expectation–Maximization (EM) algorithm and NB have been used. However, the accuracy of both algorithms for detecting rare attacks was not significantly higher as the Naïve Bayes had an accuracy of 78.06% and the accuracy of EM was 58.88%.

*Moustafa & Slay (2016)* further extended their work in 2016 and used correlation coefficient and gain ratio for feature selection in their work. Thereafter, five classification algorithms of NB, DT, ANN, LR and EM were used on the UNSW-NB15. Results showed that 85% accuracy was achieved using DT with 15.75 False Alarm Rate (FAR). This research utilized a subset of UNSW-NB15; however, detection accuracy was not satisfactory.

For detecting botnets and their tracks, *Koroniotis et al. (2017)* presented a framework using machine learning techniques on a subset of the UNSW-NB15 dataset using network flow identifiers. Four classification algorithms were used *i.e.*, Association Rule Mining (ARM), ANN, NB and DT. The results showed that the DT obtained the highest accuracy of 93.23% with a False Positive Rate (FPR) of 6.77%.

**Table 1 Summary of Existing Studies related to Network Attack Categories.**

| Reference | Dataset (complete/partial) | Algorithms | Accuracy/ FAR/FPR | Limitations |
|---|---|---|---|---|
| Moustafa & Slay (2015) | UNSW-NB15 (Partial) KDD99 Dataset | Naïve Bayes and EM Algorithm | Accuracy: Naïve Bayes–37.5% EM Algorithm: 75.80% FPR: 22.08 | This research is determining only five network attack categories. The problem of class imbalance has not been resolved. As a result, the algorithms are not performing well. |
| Moustafa & Slay (2016) | UNSW-NB15 (Partial) Dataset | Naïve Bayes, decision tree, artificial neural network, logistic regression, and expectation-maximisation | Accuracy: between 78.47% to 85.56% FAR: between 15.75% to 23.79% | In this study, data pre-processing techniques have not been implemented and the issue of class imbalance has not been resolved. |
| Koroniotis et al. (2017) | UNSW-NB15 Dataset | Naive Bayes, decision tree, association rule mining (ARM) and artificial neural network | Accuracy: between 63.97% to 93.23% FPR: between 6.77% to 36.03% | This work didn't solve the problem of class imbalance. Hence, the algorithms did not perform well to detect some network attacks. |
| Meftah, Rachidi & Assem (2019) | UNSW-NB15 Dataset | Logistic regression, gradient boost machine, and support vector machine | Accuracy: achieving a multi-classification accuracy of 86.04% | This research didn't address the class imbalance problem. |
| Kumar et al. (2020) | UNSW-NB15 (Partial) and RTNITP18 Dataset | Decision tree models (C5, CHAID, CART, QUEST) | Accuracy: 84.83% using proposed model and 90.74% using C5 model | This work has predicted only 4 out of 9 categories of the UNSW-NB15 dataset. Also, the problem of class imbalance has not been solved in this study. |
| Kasongo & Sun (2020) | UNSW-NB15 Dataset | Logistic regression, K-nearest neighbors, artificial neural network, decision tree and support vector machine | Accuracy: between 53.43% to 82.66% using multiclass classification scheme | The problem of class imbalance has not been resolved in this research. Hence, the model has not achieved good accuracy. |
| Kumar, Das & Sinha (2021) | UNSW-NB15 (Partial) Dataset | Decision tree models (C5, CHAID, CART, QUEST) | Accuracy: 88.92% using proposed model | Research has predicted only 4 types of network attack categories of the UNSW-NB15 dataset. In addition, the problem of class imbalance has not been resolved in this research. |

In 2019, Meftah, Rachidi & Assem (2019) applied a two-stage anomaly-based NIDS approach to detect network attacks. The proposed method used LR, Gradient Boost Machine (GBM) and Support Vector Machine (SVM) with the Recursive Feature Elimination (RFE) and RF feature selection techniques on a complete UNSW-NB15

dataset. The results showed that the accuracy of multi-classifiers using DT was approximately 86.04%, respectively.

*Kumar et al. (2020)* proposed an integrated calcification-based NIDS using DT models with a combination of clusters created using the k-mean algorithm and IG's feature selection technique. The research utilized only 22 features and four types of network attacks of UNSW-NB15 dataset, and the RTNITP18 dataset, which served as a test dataset to test the performance of the proposed model. The result showed an accuracy of 84.83% using the proposed model and 90.74% using the C5 model of DT.

*Kasongo & Sun (2020)* presented the NIDS approach using five classification algorithms of LR, KNN, ANN, DT and SVM in conjunction with the feature selection technique of the XGBoost algorithm. The research used the UNSW-NB15 dataset to apply binary and multiclass classification methods. Although binary classification performed well with an accuracy of 96.76% using the KNN classifier, multiclass classification didn't perform well as it achieved the highest accuracy of 82.66%.

*Kumar, Das & Sinha (2021)* proposed Unified Intrusion Detection System (UIDS) to detect normal traffic and four types of network attack categories by utilizing UNSW-NB15 dataset. The proposed UIDS model was designed with the set of rules (R) derived from various DT models including k-means clustering and IG's feature selection technique. In addition, various algorithms such as C5, Neural Network and SVM were also used to train the model. As a result, the proposed model improved with an accuracy of 88.92% over other approaches. However, other algorithms such as C5, Neural Network and SVM achieved an accuracy of 89.76%, 86.7% and 78.77%, respectively.

From a brief review of related literature as shown in Table 1, it is evident that more work needs to be done to identify the features for the families of network attacks. There is a need to determine a generic model that provides better accuracy for all the attacks presented in the dataset.

This research provides a model that determines a common subset of features. Subsequently, by using that feature subset we would be able to identify all attacks, belonging to any category with consistent accuracy. It focuses on the implementation of a generic model that provides improved classification accuracy. Moreover, there is limited research that has used the class imbalance technique to balance instances of rare attacks present in the dataset.

## PROPOSED METHODOLOGY

The framework utilizes a subset of the UNSW-NB15 dataset. It consists of two main steps. The first step involves data pre-processing, in which standardization and normalization of data are performed. Due to the high dimensional nature of the dataset, some features that are irrelevant or redundant may lead to reduce the accuracy of attack detection. To solve this problem, feature selection is used, in which only the relevant subset of features is selected to eliminate useless and noisy features from multidimensional datasets. Afterward, we have then addressed the class imbalance problem. In the next step, different classifiers are trained with relevant features to detect all categories of attack to get maximum accuracy. Finally, accuracy, precision, recall and F1-score performance

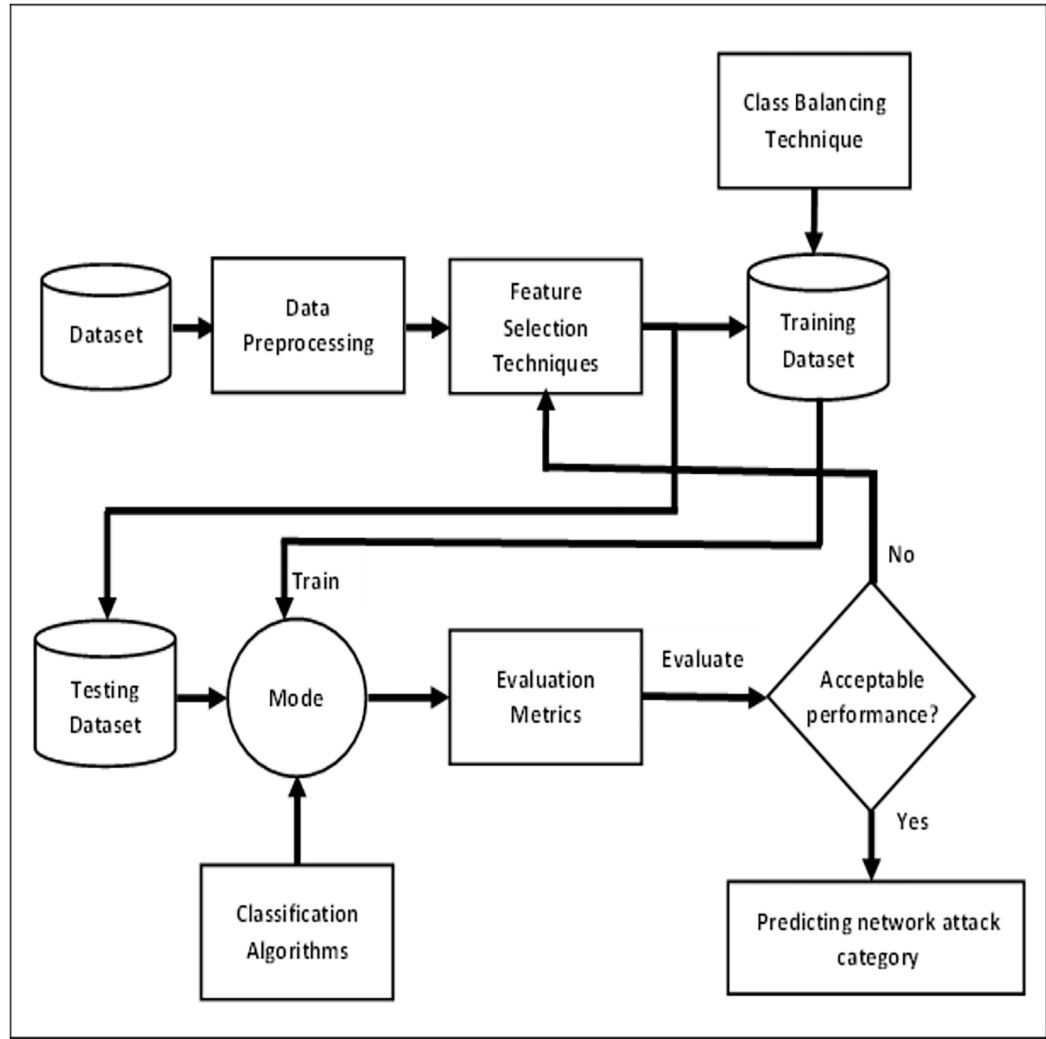

**Figure 1 Overall framework for predicting network attack categories.**

measures are used to evaluate the model. The proposed methodology that represents the overall framework is shown in Fig. 1.

## Dataset

UNSW-NB15 dataset has been created by researchers in 2015 focusing on advanced network intrusion techniques. It contains 2.5 million records with 49 features (*Dahiya & Srivastava, 2018*). There are nine different classes of attack families with two label values *i.e.*, normal or attack (*Khan et al., 2019*; *Khammassi & Krichen, 2020*) in the UNSW-NB15 dataset (*Benmessahel, Xie & Chellal, 2018*). These classes are described in Table 2.

## Dataset pre-processing

This phase involves the following steps: data standardization and data normalization.

**Table 2 Description of Network Attacks.**

| Attack family | Description |
|---|---|
| Fuzzers | These attacks attempt to crash servers on networks by inputting numerous random data, called "Fuzz", in vulnerable points of the networks |
| Analysis | These attacks perform scanning of networks *via* ports (for example: port scans, footprinting, vulnerability scans). |
| Backdoors | In this type of attacks intruder bypass normal authentication process of system portals to get illegal access into system. These attacks use malicious software's that provide remote access to the system to the attackers. |
| Denial of service (DoS) | In this type of attacks, the attackers send several illegal connection requests to generate unwanted network traffic and make network services unavailable for actual users. |
| Exploits | In these attacks vulnerable points in the operating systems are targeted and compromised. |
| Generic | This attack is a collision attack in which attackers tamper secret keys generated by using cryptographic principles. |
| Reconnaissance | These types of attacks try to find possible vulnerabilities in the computer network and then further use different exploitation techniques to exploit a compromised network. |
| Shellcode | These attacks comprise a set of instructions that are used as a payload in the exploitation of a certain network. These codes are inserted into software to compromise and remotely access a computer system. |
| Worms | These attacks are self-replicating malicious programs that exploit computer systems by duplicating themselves to the uninfected computer systems of the entire network system |

### Data standardization

As there are features with different ranges of values in the dataset we performed data standardization to convert the data from normal distribution into standard normal distribution. Therefore, after rescaling, a mean value of an attribute is equal to 0 and the resulting distribution is equal to the standard deviation. The formula to calculate a standard score (z-score) is:

$$z = \frac{(x - \mu)}{\sigma}$$

where x is the data sample, μ is the mean and σ is the standard deviation (*Xiao et al., 2019*).

### Data normalization

In data normalization, the value of each continuous attribute is scaled between 0 and 1 such that the result of attributes does not dominate each other (*Gupta et al., 2016*). In this research, the normalizer class of Python has been used. This class enables the normalization of a particular dataset.

### Feature selection

Feature selection is a technique that is used to select features that mostly correlate and contribute to the target variable of the dataset (*Aljawarneh, Aldwairi & Yassein, 2017*). In this research, feature selection is done using Correlation Attribute Evaluation (CA), Information Gain (IG) and Principal Component Analysis (PCA). CA measures the relationship between each feature with the target variable and select only those relative features that have moderately higher positive or negative values, *i.e.*, closer to 1 or −1 (*Sugianela & Ahmad, 2020*). While IG feature selection technique is used to determine

| Table 3 Class distribution in dataset. | |
|---|---|
| **Class Distribution** | **Instance count (%)** |
| Normal | 32.10 |
| Generic | 22.80 |
| Exploits | 19.06 |
| Fuzzers | 10.32 |
| DoS | 7.08 |
| Reconnaissance | 5.85 |
| Analysis | 1.12 |
| Backdoor | 0.93 |
| Shellcode | 0.65 |
| Worms | 0.06 |

relevant features and minimizing noise caused by unrelated features. These relevant features are calculated from the entropy matrix which measures the uncertainty of the dataset (*Kurniabudi et al., 2020*). Through Principal Component Analysis, the size of large datasets is reduced by retaining the relevant features that depend on the target class (*Kumar, Glisson & Benton, 2020*).

The above-mentioned feature selection techniques help to train the model correctly with only the relevant features that accurately predict the target class.

## Class imbalance

The UNSW-NB15 dataset is highly imbalanced not only because the number of normal traffic instances is much higher than different attack categories, but also because the different categories of attack instances are not equal in distribution. This problem is known as "Class Imbalance".

Table 3 depicts the distribution of nine categories of attack and normal instances in the training dataset. The attack categories such as Analysis, Backdoor, Shellcode and Worms have very few instances. This highly imbalanced nature of the dataset causes problems in training machine learning algorithms for accurately predicting cyber-attacks. To address the class imbalance issue, this research uses SMOTE. SMOTE synthesizes instances of minority classes to balance all the classes in the dataset (*Laureano, Sison & Medina, 2019*). Table 4 shows the instance percentages in each class after applying SMOTE.

## Classification algorithms

Five classification algorithms, that is, RF, DT, LR, KNN and ANN were employed to train the model.

### Random forest

Random Forest is an ensemble classifier that is used for improving classification results. It comprises multiple Decision Trees. In comparison with other classifiers, RF provides lower

**Table 4 Class distribution in dataset after applying SMOTE.**

| Class Distribution | Instance count (%) |
| --- | --- |
| Normal | 22.13 |
| Generic | 18.31 |
| Exploits | 15.72 |
| Fuzzers | 10.05 |
| DoS | 7.11 |
| Reconnaissance | 6.62 |
| Analysis | 5.21 |
| Backdoor | 5.08 |
| Shellcode | 4.95 |
| Worms | 4.76 |

classification errors. Randomization is applied for the selection of the best nodes for splitting when creating separate trees in RF (*Jiang et al., 2018*).

### Decision tree

In the Decision Tree algorithm, the attributes are tested on internal nodes, the outcomes of the tests are represented by branches, and leaf nodes hold labels of the classes (*Afraei, Shahriar & Madani, 2019*). Attribute selection methods are used for identifying nodes. Those selected attributes minimize the information that is required for tuple classification in the resulted partition. Hence, reflecting the minimum uncertainty or impurity in those partitions. Therefore, minimizing the projected number of tests required for tuple classification. In this research, ID3 algorithms utilize entropy class to determine which attributes should be queried on, at every node of those decision trees.

### Logistic Regression

Logistic Regression is a probabilistic classification model. It casts the problem into a generalized linear regression form. It has a sigmoid curve. The equation of the sigmoid function or logistic function is:

$$S(x) = \frac{e^x}{1 + e^x}$$

This function is used for mapping values to probabilities. It works by mapping real values to other values between 0 and 1 (*Kyurkchiev & Markov, 2016*).

### K-nearest neighbors

In K-Nearest Neighbors, a new data point is attached with the data points in the training set and based on that attachment, a value is assigned to that new data point. This uses feature similarity for prediction. In KNN Euclidean, Manhattan or Hamming distance are used for calculating the distance between a test data and each record of training data (*Jain, Jain & Vishwakarma, 2020*). Afterward, according to the value of distance, the rows are sorted. From those rows, K rows from the top are selected. Based on the most frequent classes of those rows, classes to the test points are assigned.

### Artificial neural network

In the Artificial Neural Network algorithm, there are three layers that consist of computational units called neurons. These layers are input, output and hidden layers. The number of neurons in these layers depends on the features of the dataset and classes which have to be detected and chosen with different techniques. Different types of activation functions are used in the ANN algorithm for calculating the weighted sum of the connections between neurons. This algorithm has biases in the hidden layer and an output layer which are adjusted to reduce errors and improve accuracy in training and testing the model (*Andropov et al., 2017*).

## Evaluation metrics

A confusion matrix is used for the comparison of the performance of machine learning algorithms. This matrix is used for the creation of different metrics by the combination of the values of True Negative (TN), True Positive (TP), False Negative (FN) and False Positive (FP) (*Tripathy, Agrawal & Rath, 2016*). Below are some of the performance measures to evaluate models by the use of the confusion matrix.

Accuracy shows the correctness or closeness of the approximated value to the actual or true value of the model which means a portion of the total samples that are classified correctly (*Lin, Ye & Xu, 2019*). The following formula is used to calculate the accuracy of the model:

$$Accuracy = \frac{TP + TN}{TP + TN + FN + FP}$$

Precision shows which portion of relevant instances is actually positive among the selected instances (*Roy & Cheung, 2018*). The following formula is used to calculate precision:

$$Precision = \frac{TP}{TP + FP}$$

Recall or True Positive Rate (TPR) calculates the fraction of actual positives that are correctly identified (*Ludwig, 2017*). The formula used to find recall is:

$$Recall = \frac{TP}{TP + FN}$$

F1-score is interpreted as the harmonic mean of precision and recall means it combines the weighted average of precision and recall (*Javaid et al., 2016*). The following formula is used to calculate F1-score:

$$F1 = 2 * \frac{Precision * Recall}{Precision + Recall}$$

**Table 5 Results of classification models without feature selection.**

| Method | Accuracy% | Precision% | Recall% | F1-Score% |
|---|---|---|---|---|
| Random forest | 89.29 | 76.9 | 72.6 | 74.1 |
| Decision tree | 88.29 | 70.5 | 72.7 | 71.4 |
| Logistic regression | 82.07 | 50.5 | 42.6 | 42.3 |
| K-nearest neighbor | 82.37 | 52.5 | 47.6 | 49.1 |
| Artificial neural network | 84.80 | 61.0 | 53.2 | 54.5 |

# EXPERIMENT AND RESULT ANALYSIS

Following the methodology depicted in Fig. 1, experimental setup is established. In this research, a sample of 80,000 instances is randomly selected from the UNSW-NB15 dataset. Initially, data standardization and normalization have been performed to rescale data values of the dataset and then three feature selection techniques are applied to select the most relevant features. Afterward, the class imbalance problem is resolved using SMOTE. Lastly, five classification algorithms *i.e.*, RF, DT, LR, KNN and ANN are used to classify between the attack categories and normal traffic.

## Performance analysis of classification models without feature selection

Table 5 summarizes the results of classification models without feature selection, in which RF achieved the highest accuracy, precision, recall, and F1-score values approximately 89.29%, 76.9%, 72.6% and 74.1% respectively. Whereas, LR obtained the lowest accuracy, precision, recall, and F1-score values of approximately 82.07%, 50.5%, 42.6% and 42.3%. DT achieved an accuracy of 88.29% with 70.5% precision, 72.7% recall and 71.4% F1-score. KNN also recorded the same accuracy as LR but with better precision, recall and F1-score results. Whereas, ANN recorded average performance with accuracy, precision, recall, and F1-score values of approximately 84.80%, 61.0%, 53.2% and 54.5%.

## Performance analysis of classification models with feature selection

Three feature selection techniques *i.e.*, CA, IG and PCA are used in this research.

The following 35 features are selected by applying IG: sttl, ct_state_ttl, ct_flw_http_mthd, sbytes, id, smean, sload, dur, sinpkt, rate, proto, ct_dst_src_ltm, service, dbytes, sjit, ct_srv_dst, dload, dinpkt, dmean, ct_srv_src, synack, tcprtt, ct_dst_sport_ltm, djit, ct_src_dport_ltm, dtcpb, stcpb, spkts, dloss, ct_dst_ltm, ackdat, label, dpkts, ct_src_ltm, sloss.

After applying CA method, 24 features have been achieved from the set of 49 features: id, ct_dst_sport_ltm, ct_dst_src_ltm, ct_src_dport_ltm, sttl, ct_srv_dst, ct_srv_src, ct_dst_ltm, ct_src_ltm, ct_state_ttl, state, swin, dwin, proto, service, rate, dttl, stcpb, dtcpb, dmean, dload, tcprtt, ackdat, synack.

By applying PCA, 151 subsets from the set of 49 features were resulted, out of which 10 subsets with 80% results were selected. After tremendous analysis and evaluation of these 10 subsets, 24 features have been extracted. Afterward, five classifiers were trained by

**Table 6 Results of classification models with feature selection.**

| Method | Feature selection techniques | Accuracy% | Precision% | Recall% | F1-Score% |
|---|---|---|---|---|---|
| Random forest | Information gain attribute | 89.5 | 76.8 | 72.3 | 73.7 |
|  | Correlation attribute | 86.3 | 72.2 | 59.6 | 61.9 |
|  | Principal component analysis | 89.3 | 77.3 | 70.8 | 73.1 |
| Decision tree | Information gain attribute | 88.5 | 69.6 | 72.0 | 70.7 |
|  | Correlation attribute | 84.8 | 63.3 | 60.9 | 61.9 |
|  | Principal component analysis | 88.4 | 70.9 | 67.3 | 69.3 |
| Logistic regression | Information gain attribute | 82.2 | 51.2 | 42.3 | 41.9 |
|  | Correlation attribute | 74.5 | 38.5 | 36.0 | 35.4 |
|  | Principal component analysis | 80.4 | 51.3 | 40.6 | 40.0 |
| K-nearest neighbor | Information gain attribute | 82.7 | 54.0 | 48.1 | 49.6 |
|  | Correlation attribute | 76.8 | 46.3 | 42.6 | 43.5 |
|  | Principal component analysis | 84.0 | 57.8 | 51.3 | 53.3 |
| Artificial neural network | Information gain attribute | 85.7 | 60.6 | 54.6 | 54.2 |
|  | Correlation attribute | 80.3 | 50.1 | 44.1 | 44.7 |
|  | Principal component analysis | 85.2 | 61.2 | 52.4 | 54.4 |

using these features: id, dur, dwin, proto, djit, swin, smean, state, service, ct_src_dport_ltm, dbytes, ct_dst_ltm, ct_dst_sport_ltm, ct_src_ltm, dloss, ct_flw_http_mthd, ct_srv_dst, dpkts, sttl, dmean, spkts, sbytes, sloss and sinpkt.

After training the classifiers with the above features, the results showing in Table 6 were obtained.

It is observed using IG technique, that the RF classifier achieved the highest accuracy of 89.5% approx. with precision rate (76.8%), recall (72.3%) and F1-score (73.7%). In contrast to other classifiers, LR and KNN didn't perform well with IG as their recall and F1-score have below 50% scores. There is no much difference between the accuracy of RF and DT classifiers as both give almost the same accuracy, recall and F1-score measures using IG technique. The only difference is in the precision rate as RF achieved 76.8% and DT scored 69.6% precision value.

It is observed that the accuracy of all the classifiers decreased when the model is trained using the CA technique. RF classifier achieved the highest accuracy of 86.3% but with low precision, recall and F1-score measures. The accuracy of DT and ANN classifiers are approximately the same as the RF classifier with a minor difference of 2% to 5%. However, ANN classifier has very low-performance measures as compared to RF and DT. Also, LR and KNN have the lowest accuracy measures with poor performance metrics.

It is observed using PCA feature selection technique, that RF classifier obtained the highest accuracy of 89.3% with precision (77.3%), recall (70.8%) and F1-score (73.1%) rates. All the classifiers achieved the accuracy in between 80% to 89% but with low performance measures as compared to IG feature selection technique. LR recorded the lowest recall rate with 40.6% and F1-score with 40%.

After evaluation of the performance of three feature selection methods, it was observed that the feature selection technique of IG and PCA performed well as compared to CA.

**Table 7 Results of classification models with feature selection after handling class imbalance.**

| Method | Feature selection techniques | Accuracy% | Precision% | Recall% | F1-Score% |
|---|---|---|---|---|---|
| Random forest | Information gain attribute | 95.0 | 94.7 | 95.7 | 95.1 |
| | Correlation attribute | 93.5 | 93.0 | 94.2 | 93.5 |
| | Principal component analysis | 95.1 | 94.8 | 95.7 | 95.1 |
| Decision tree | Information gain attribute | 94.5 | 94.1 | 95.2 | 94.7 |
| | Correlation attribute | 92.6 | 91.8 | 93.4 | 92.6 |
| | Principal component analysis | 94.7 | 94.4 | 95.4 | 94.8 |
| Logistic regression | Information gain attribute | 69.4 | 61.0 | 59.4 | 56.2 |
| | Correlation attribute | 62.0 | 48.1 | 50.6 | 45.8 |
| | Principal component analysis | 68.2 | 57.8 | 58.0 | 54.8 |
| K-nearest neighbor | Information gain attribute | 82.7 | 79.4 | 82.6 | 80.6 |
| | Correlation attribute | 78.4 | 74.3 | 78.3 | 75.8 |
| | Principal component analysis | 84.7 | 82.2 | 85.1 | 83.1 |
| Artificial neural network | Information gain attribute | 77.3 | 75.0 | 70.5 | 71.0 |
| | Correlation attribute | 71.7 | 69.2 | 62.1 | 63.5 |
| | Principal component analysis | 77.6 | 76.2 | 70.6 | 71.5 |

RF and DT classifiers approximately achieved the same accuracy between 88% to 89% when trained with IG and PCA. However, for precision, recall and F1-score measures, these classifiers showed average scores. Therefore, it is concluded that, no major changes have been observed in the results after applying feature selection techniques as classifiers achieved almost same accuracy before feature selection.

## Performance analysis of classification models by handling imbalanced data

To handle imbalanced data, SMOTE technique has been applied in this research to adjust the class distribution of dataset and increase the instances of minority classes of those network attacks that has lower instances. After handling imbalance data, the results showing in Table 7 were obtained.

By using IG feature selection technique after applying SMOTE, it is observed that the RF classifier achieved highest accuracy of 95.0% with highest precision rate (94.7%), recall (95.7%) and F1-score (95.1%). Also, the accuracy of DT is 94.5%, almost nearest to the RF. The accuracy of both algorithms increased after handling imbalanced classes *i.e.*, from 89.5% to 95.0% in RF and 88.5% to 94.5% in DT. Whereas, after applying SMOTE, LR and ANN didn't perform well as their accuracies were decreased from 82.2% to 69.4% in LR and 85.7% to 77.3% in ANN using IG method. The accuracy of KNN is almost the same using all three feature selection techniques but with good precision, recall and F1-score measures.

By using CA feature selection technique after applying SMOTE, it is noticed that RF and DT classifiers achieved highest accuracy in between 92.6% to 93.5% with above 90% precision, recall and F1-score measures. The accuracy of both the algorithms increased

after applying SMOTE. Also, a minor change occurred in KNN as their accuracy is improved from 76.8% to 78.4% after handing imbalanced classes. However, when comes to LR and ANN, both algorithms did not perform well with class balance as their accuracies have been decreased *i.e.*, in LR, from 74.5% to 62.0% and in ANN, from 80.3% to 71.7%.

By using PCA after applying SMOTE, it is noticed that the RF classifier achieved the highest accuracy of 95.1% with a precision rate of 94.8%, recall of 95.7% and F1-score of 95.1%. DT classifier achieved the accuracy of 94.7%, which is almost nearest to the accuracy of RF. After applying SMOTE, there is no change in the results of KNN using PCA method. However, the accuracy of LR and ANN decreased from 80.4% to 68.2% in LR and 85.2% to 77.6% in ANN but with increased precision, recall and F1-score measures.

## Overall performance evaluation of classification models after handling class imbalance

After handling class balancing by using SMOTE, it is concluded that RF classifier performed well with good results up to 95.1% by using PCA feature selection technique. Also, it is noticed that class balancing did not impact on LR and ANN classifiers as their accuracy decreased after handling minority classes.

### *Confusion metrics of best performed classifier: random forest*

After analysis of the five classification models, it is observed that RF scheme provided the highest accuracy. On the basis of which, the confusion matrix of RF classification model is analyzed to observe the attack prediction accuracy of the nine categories of attacks separately.

In Fig. 2, it is depicted that all the normal traffic instances were identified correctly by RF (*i.e.*, it had 100% accuracy). In attack categories, all the instances of Backdoor, Shellcode and Worms were also identified correctly showing 100 prediction accuracy. Whereas, 1,759 out of 1,763 instances of Analysis attack (*i.e.*, 99.77% accuracy), 2,341 out of 2,534 instances of Fuzzers (*i.e.*, 92.38% accuracy), 5,461 out of 5,545 instances of Generic (*i.e.*, 98.49% accuracy), 2,151 out of 2,357 instances of Reconnaissance (*i.e.*, 91.26% accuracy) were identified correctly.

## DISCUSSION

The research proposed a framework that predicts a variety of network attack categories using supervised machine learning algorithms. The dataset used in this study is the UNSW-NB15 dataset, a relatively new and containing a large amount of network traffic data, with nine types of network attack categories.

The proposed framework implies five machine learning algorithms in conjunction with pre-processing techniques, different methods of feature selection and SMOTE. After training, the results of the classifiers shown in Tables 6 and 7 were obtained.

Compared to previous studies, as shown in Table 1, our model performed well with the highest accuracy of 95.1% using RF classifier with 24 features selected by PCA after applying SMOTE. DT classifier has also performed well with accuracy between 92.6% to

| | Predicted | 0 | 1 | 2 | 3 | 4 | 5 | 6 | 7 | 8 | 9 |
|---|---|---|---|---|---|---|---|---|---|---|---|
| | True | | | | | | | | | | |
| Analysis | 0 | 1759 | 0 | 2 | 2 | 0 | 0 | 0 | 0 | 0 | 0 |
| Backdoor | 1 | 0 | 1762 | 0 | 0 | 0 | 0 | 0 | 0 | 0 | 0 |
| DoS | 2 | 39 | 33 | 3038 | 322 | 60 | 6 | 0 | 36 | 2 | 0 |
| Exploits | 3 | 51 | 33 | 424 | 5626 | 78 | 8 | 0 | 130 | 14 | 3 |
| Fuzzers | 4 | 10 | 13 | 56 | 78 | 2341 | 0 | 0 | 11 | 21 | 0 |
| Generic | 5 | 1 | 3 | 22 | 44 | 9 | 5461 | 0 | 0 | 0 | 0 |
| Normal | 6 | 0 | 0 | 0 | 0 | 0 | 0 | 7453 | 0 | 0 | 0 |
| Reconnaissance | 7 | 6 | 18 | 64 | 102 | 9 | 2 | 0 | 2151 | 0 | 0 |
| Shell code | 8 | 0 | 0 | 0 | 0 | 0 | 0 | 0 | 0 | 1651 | 0 |
| Worm | 9 | 0 | 0 | 0 | 0 | 0 | 0 | 0 | 0 | 0 | 1846 |

**Figure 2 Confusion matrix of random forest classifier.**

94.7% using different feature selection techniques. Existing studies that summarized in Table 1, achieved less than 90% accuracy, except for the research proposed by *Kumar et al. (2020)* and *Koroniotis et al. (2017)*. *Kumar et al. (2020)* showed 90.74% accuracy using the C5 model of DT in conjunction with the IG feature selection technique. While the model proposed by *Koroniotis et al. (2017)* achieved 93.23% accuracy using DT classifier. In addition, none of the studies listed in Table 1 have resolved the class imbalance problem of the UNSW-NB15 dataset as there are many studies (*Al-Daweri et al., 2020*; *Ahmad et al., 2021*; *Bagui & Li, 2021*; *Dlamini & Fahim, 2021*) that have highlighted this issue. We addressed the class imbalance problem by applying SMOTE that improved the performance of the classifiers and achieved good results.

## CONCLUSIONS

This paper presents a framework for network intrusion detection. The performance of the proposed framework has been analyzed and evaluated on the UNSW-NB15 dataset. The proposed framework uses different pre-processing techniques that includes data standardization and normalization, feature selection techniques and class balancing methods. The usability of the selected features along with using data standardization and normalization techniques is analyzed by applying them on five different classification models. The results showed that the features selected by PCA contributed much to improve accuracy than other methods. For improving the accuracy of the classification models, the class imbalance problem is also addressed which increased the framework performance with high margin. In can be concluded on the basis of evaluation results that both RF and DT classifiers performed well over the UNSW-NB15 dataset in terms of accuracy, precision, recall, and F1-score metrics. It can also be concluded that major issue in UNSW-NB15 dataset is not only the presence of highly correlated features but also the

class imbalance problem of the dataset. Therefore, we used a novel combination of different pre-processing techniques in order to resolve all the underlying issues of the dataset and developed a fast and efficient network security intrusion detection system.

### Funding
The authors received no funding for this work.

### Competing Interests
The authors declare that they have no competing interests.

### Author Contributions
- Hafiza Anisa Ahmed conceived and designed the experiments, performed the experiments, performed the computation work, prepared figures and/or tables, and approved the final draft.
- Anum Hameed conceived and designed the experiments, performed the experiments, authored or reviewed drafts of the paper, and approved the final draft.
- Narmeen Zakaria Bawany analyzed the data, authored or reviewed drafts of the paper, and approved the final draft.

### Patent Disclosures
The following patent dependencies were disclosed by the authors:
The UNSW-NB15 Dataset is available at: https://research.unsw.edu.au/projects/unsw-nb15-dataset.

### Data Availability
The code is available at GitHub: https://github.com/hafizaanisa29/Network_Attack_Research_CodeFiles.

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
