# Peer review of "Network intrusion detection using oversampling technique and machine learning algorithms"

_PeerJ Computer Science, doi:10.7717/peerj-cs.820_

## Round 0.1 · original submission · Major Revisions

Please see the review reports and address the highlighted concerns.

Reviewer 1 ·

Basic reporting

1.The author claims in lines 70-71 that the proposed model performs better than previous related research, but there is no specific explanation and comparison in the paper, which is hard to convince. I suggest that the experiment is not just to compare the performance differences of existing Methodologies, but to compare the superiority of the detection framework over the previous research when facing the same dataset.
2.Lines 72-73 indicate that the data preprocessing technology mentioned in this paper is novel and has not been employed. However, there are abundant researches on data standardization and normalization technology and SMOTE technology for dealing with imbalances. So this statement is debatable. Readers are more concerned about the innovation of this work. Therefore, the author should not only declare the novelty of the work but also discuss and prove it in detail. This will help readers grasp the value of the work.

Experimental design

3.There are some errors in the presentation of data, tables and conclusions:
1)Since the IG method is used, the description from 89.5% to 95.1% in RF in line 358 should be modified from 89.5% to 95% in RF.
2)The decrease in ANN accuracy is incorrectly stated in lines 370 to 371 because 76.8% is the accuracy of KNN. The ANN accuracy decreases from 80.3% to 71.7%.
3)In lines 377 to 378 the conclusion is wrong, the accuracy of KNN increases from 84.0% to 84.7% after using PCA, which does not reflect the decrease in accuracy with PCA, and we note that the accuracy of ANN drops exactly 85.2% to 77.6% again after using PCA.
In line 385 the conclusion is wrong, class balancing does not affect Logistic Regression and Artificial Neural Networks, not Decision Tree and Artificial Neural Networks as described in the paper.

Validity of the findings

no comment

Additional comments

no comment

·

Basic reporting

• In general, the abstract should be about 240/260 words in length and up to the 500 words limit. It can contain short statements summarising the project scope, research significance/motivation/problem statement, aim, objectives, the method and techniques used to work towards these objectives, and the results achieved and conclusions made. It should give a reader sufficient information to decide whether or not to read the rest of your paper. Make sure to revise your abstract to address most of the above requirements, especially your motivation and aims.
• The abstract keywords are not presented.
• The uncommon abbreviations should be spelt out at first use only. EM algorithm in line 107 is not abbreviated, and SMOTE is abbreviated twice in lines 87 & 199. Therefore, the English language should be improved, I suggest you have a colleague who is fluent in English and familiar with the subject matter review your manuscript.
• Make sure all the equations are in the middle of the line.
• In line 318 the table name error.
• In line 155 it’s F1-score, and not f1-score.
• Line 101 – 103, the reference style in table 1 is not as the main text, also, there is no reference to what you cited in this table. Also, you should add more research from 2020 and 2021 in this table.

Experimental design

• In lines 66-67, you need to clarify why you used only the UNSW-NB15 dataset and update this section as in 2021 more than 24 open access papers mention the UNSW-NB15 dataset. https://paperswithcode.com/dataset/unsw-nb15
• In line 71 you mentioned the previous research, and you claim that your result achieves better accuracy than the previous research. You should cite them here.
• In lines 79-79 you should provide detailed information about your experimental setup.
• A methodology diagram in the methodology section will benefit this section.
• In line 316, the label has been extracted as a feature, can you clarify why?
• In lines 184-188 More details and discussion are required in this section.
• The percentage in table 3 is not accurate if you consider all datasets and not just the training datasets. You can refer to Table VIII in the reference given below for more details. N. Moustafa and J. Slay, "UNSW-NB15: comprehensive data set for network intrusion detection systems (UNSW-NB15 network data set)," 2015 Military Communications and Information Systems Conference (MilCIS), 2015, pp. 1-6, doi: 10.1109/MilCIS.2015.7348942.

Validity of the findings

• In this paper, the authors used the SMOTE technique and feature selection techniques to address the class imbalance issue for some attack types that have very few instances on the UNSW-NB15 Dataset. They managed to achieve a better result when applying both techniques. However, the authors haven’t validated their results with the current research results. A Validation section should be included in this paper.

Additional comments

• Only one dataset has been tested with the proposal framework, I recommend adding more than one dataset to evaluate their framework.
• I noticed that the raw data and the code are not submitted
• However, I do think the authors could further improve the papers by addressing the above comments, and adding a discussion section at the end of this paper will make it even more readable and comprehensive.

---

## Round 0.2 · accepted · Accept

All of the reviewers' concerns have been addressed. This paper can be accepted.

Reviewer 1 ·

Basic reporting

The author has answered my concerns and I think it can be published.

Experimental design

The author has answered my concerns and I think it can be published.

Validity of the findings

The author has answered my concerns and I think it can be published.

Additional comments

The author has answered my concerns and I think it can be published.

·

Basic reporting

I recommend accepting this paper; it is much improved from the previous version. It's now relatively easy to read from beginning to end, and the technical content of the paper has been settled, so I have no additional technical comments on this one

Experimental design

.

Validity of the findings

.